# Exploring the interplay between *Porphyromonas gingivalis* KGP gingipain, herpes virus MicroRNA-6, and Icp4 transcript in periodontitis: Computational and clinical insights

**Pradeep Kumar Yadalam[1], Anila Neelakandan[2], Rex Arunraj[3], Raghavendra Vamsi Anegundi[1], Carlos M. Ardila[4,5]***

1 Department of Periodontics, Saveetha Dental College, Saveetha Institute of Medical and technology sciences, SIMATS, Saveetha University, Chennai, Tamil Nadu, India, 2 Clover Medical Center, Dubai, United Arab Emirates, 3 Department of Genetic Engineering, School of Bioengineering, SRM Institute of Science and Technology, Kanchipuram, Tamil Nadu, India, 4 Basic Sciences Department, Faculty of Dentistry, Universidad de Antioquia U de A, Medellín, Colombia, 5 Biomedical Stomatology Research Group, Universidad de Antioquia U de A, Medellín, Colombia

* martin.ardila@udea.edu.co

**Data Availability Statement:** "All relevant data are within the paper and its Supporting information files."

## Abstract

### Background

*Porphyromonas gingivalis*, a major pathogen in periodontitis, produces KGP (Lys-gingipain), a cysteine protease that enhances bacterial virulence by promoting tissue invasion and immune evasion. Recent studies highlight microRNAs' role in viral latency, potentially affecting lytic replication through host mechanisms. Herpes virus (HSV) establishes latency via interactions between microRNA-6 (miRH-6) and the ICP4 transcription factor in neural ganglia. This suggests a potential link between periodontitis and HSV-induced latency. This study aims to identify and validate the insilico inhibitory interaction of *P. gingivalis* KGP with ICP4 transcripts and correlate the presence of viral latency-associated transcript micro-RNA-6 with periodontitis.

### Methods

Computational docking analysis was performed to investigate the potential interaction between ICP4 and KGP gingipain. The binding energy and RMSD ligand values were calculated to determine the interaction's strength. Ten patients with recurrent clinical attachment loss despite conventional therapy were included in the clinical study. Subgingival tissue samples were collected post-phase I therapy, and HSV microRNA-6 presence was detected via polymerase chain reaction and confirmed through gel electrophoresis.

### Results

Computational docking identified the ICP4-KGP gingipain complex with the lowest binding energy (-288.29 kJ mol^1) and an RMSD ligand of 1.5 Angstroms, indicating

**Funding:** The author(s) received no specific funding for this work.

**Competing interests:** The authors have declared that no competing interests exist.

strong interaction potential. Gel electrophoresis confirmed miRH-6 presence in all samples.

## Conclusion

The identification of miRNA-6 in periodontitis patients and the strong interaction potential between *P. gingivalis* KGP gingipain and ICP4 transcripts indicate a possible link between bacterial virulence factors and viral latency dynamics in periodontal tissues. These results highlight the complex interplay between oral pathogens, viral microRNAs, and host immune responses in periodontitis.

## 1. Introduction

Periodontitis is a chronic inflammatory disease with a complex etiology. It manifests clinically as the pathological destruction of the periodontal ligament and alveolar bone. Specific bacterial species, particularly *Porphyromonas gingivalis* and herpesviruses, are recognized as key periodontal pathogens. These pathogens can synergistically interact, potentially contributing to the development and progression of severe periodontitis [1]. Periodontitis development likely involves a synergistic interaction between periodontal herpesviruses and specific bacterial species. Herpesviruses may promote bacterial colonization and growth within the periodontium [2].

Conversely, bacterial factors might reactivate latent herpesviruses. This two-way communication could further impede the host's immune response against bacteria. Elucidating these intricate interactions holds promise for novel therapeutic targets to prevent periodontitis and potentially associated systemic diseases [2]. Despite effective therapy and patient compliance, some individuals with periodontitis continue to experience attachment loss [3]. This suggests the presence of hidden reservoirs of microbial infection or the emergence of opportunistic pathogens following standard treatment.

Additionally, unknown host factors may hinder the effectiveness of traditional periodontal therapies. While past research primarily focused on identifying bacteria responsible for periodontitis, recent studies highlight the potential involvement of specific herpesviruses in disease initiation and progression [4, 5]. This shift in focus opens avenues for a more comprehensive understanding of periodontitis and developing improved treatment strategies.

Herpesviruses, such as Herpes Simplex Virus-1 (HSV-1), are known to establish long-term latent infections within their hosts [6]. During reactivation from latency in sensory ganglion neurons, HSV-1 can cause lesions on surrounding mucosal surfaces. Notably, both latent and lytic (actively replicating) herpesviruses express microRNAs (miRNAs) [1]. Several studies demonstrate that viral miRNAs regulate the host's gene expression patterns (transcriptome) [1]. Elevated herpesvirus-derived miRNAs in the gingival tissues may contribute to a weakened cellular and immune response, potentially leading to periodontal disease.

Within the context of HSV latency, miR-H6 is a specific viral miRNA associated with maintaining latency and activating productive viral replication [7, 8]. This differs from LAT, another viral transcript found in higher numbers within nerve ganglia in conditions like trigeminal neuralgia and infections of the eye and oral tissues. Viral miRNAs function as regulators of gene expression, manipulating both viral and host genes to benefit the virus. Consequently, miRNAs play a critical role in the pathogenesis of viral diseases and the complex interactions between viruses and their hosts [1].

Immediate Early Protein 4 (ICP4), expressed by Human Cytomegalovirus (HCMV) during the early stages of infection, plays a vital role in regulating viral gene expression and DNA replication [9]. This protein interacts with various cellular proteins, manipulating cellular processes to facilitate viral replication and evade the host's immune system [9].

In minimal amounts, keystone pathogens like *P. gingivalis* lead to dysbiotic bacteria colonization [10]. Thus, treated periodontitis, especially in some susceptible individuals, can activate the HSV latent virus residing in the sensory nerves of the periodontium. KGP is a protein produced by these bacteria that interacts with the latent associated herpes virus (LAHV) and microRNA (MIRH-6) to increase the replication and expression of the ICP4 transcript. ICP4 is a viral protein essential for the replication of herpes simplex virus type 1 (HSV-1). Therefore, the interaction between KGP, LAHV, and MIRH-6 can increase the proliferation of HSV-1 in the mouth, leading to oral inflammation and disease [11].

The presence of latent HSV, which *P. gingivalis* can reactivate, can lead to changes in the microbial profile in recurrent cases. The established latency of HSV is due to interactions between the miRH-6 transcription factor and ICP4 in neural ganglia [12]. To our knowledge, this is the first study to demonstrate a connection between periodontitis and HSV latency. We hypothesize that the interaction between the MIRH-6-ICP4 transcription factor and *P. gingivalis* gingipain KGP leads to the deactivation of HSV micro-RNA six and the development of viral-induced periodontitis.

This study aims to identify and validate the insilico inhibitory interaction of *P. gingivalis* KGP with ICP4 transcripts and to correlate the presence of viral latency-associated micro-RNA 6 with periodontitis. We will focus on a proof-of-concept insilico and clinical study to support our hypothesis.

## 2. Materials and methods

### 2.1. Insilico validation of *P. gingivalis* -ICP4 interaction

The interactions between the MIRH-6-ICP4 transcription factors are essential for maintaining latency in the nerve ganglia of the periodontium. To confirm the interaction of *P. gingivalis* gingipain KGP and its inhibition of ICP4, protein-protein docking of MIRH-6-induced latency is required [13].

### 2.2. Protein-protein docking

Lysine gingipains (KGP) from *P. gingivalis* facilitate epithelial cell invasion. Molecular docking of the KGP lysine gingipain with ICP4 (protein-peptide complex) was conducted using the Hdock server [13].

### 2.3. Molecular dynamics simulation

The DESMOND (version 14) molecular dynamics package assessed the stability of protein-protein interactions. Molecular dynamics simulations were conducted for 100 nanoseconds using Schrödinger LLC's Desmond software, which employs Newton's classical equations of motion to simulate atomic movements over time [14].

The Maestro Protein Preparation Wizard was utilized to optimize the receptor-ligand complex, and the System Builder tool was employed to construct all systems. The solvent model employed was TIP3P in an orthorhombic box configuration. The OPLS 2005 force field was applied, along with counter ions, to neutralize the models. The NPT ensemble was used with 0.15 M sodium chloride (NaCl) at 300 K and 1 atm pressure.

The stability of the simulations was evaluated by comparing the root mean square deviation (RMSD) of the protein and ligand every 100 ps.

## 2.4 Clinical validation study

A sample of ten patients was selected from a pool of individuals who had undergone treatment for periodontitis but continued to exhibit clinical attachment loss despite receiving standard periodontal therapy (non-surgical therapy followed by surgical therapy). These patients were chosen based on the prevalence of the condition. It was determined that each of these ten patients had one subgingival tissue sample obtained from them.

**Inclusion criteria.**

- Periodontal disease with a history of periodontitis and treatment for periodontitis but continued to exhibit clinical attachment loss [15].

- Adherence to maintenance and hygiene recommendations [15].

- Presence of at least two active sites (bleeding on probing or probing pocket depth $\geq 5$ mm) in recurrent sites

**Exclusion criteria.**

- Pregnancy

- Diabetes

- Blood disorders

- Inadequate conventional care

Given the exploratory nature of this study to identify the existence of HSV microRNA, a statistical power analysis was not conducted. The participants were fully informed about the purpose, risks, and benefits of the study, and their rights as participants. They signed a written informed consent form, which was obtained separately from each participant. Moreover, ethical approval was granted by Saveetha Dental College & Hospitals on 01-28-2022 (Ref No. 22-perio-317).

Following periodontal therapy, these ten patients were monitored for two years (01-30-2022, to 01-30-2024). During this period, each patient reported at least one instance of reactivation, evidenced by the presence of sites with probing depths and clinical attachment loss exceeding 6 mm. Patient recruitment took place at Saveetha Dental College & Hospitals, India.

Tissue samples were collected from the deepest pocket using a curette (Pocket Lining) and immediately frozen at -80°C for subsequent PCR analysis. Primers specific for miR-H6 were designed as follows:

- Forward Primer: `CCGGAGGGTGGAAGGCAG`

- Reverse Primer: `GATGGAAGGACGGGAAGTG`non s

The tissue samples were subsequently used for RNA isolation. After isolation, RNA was reverse-transcribed to generate complementary DNA (cDNA). The cDNA was then subjected to reverse transcription PCR (RT-PCR), which included an initial denaturation step at 95°C followed by multiple amplification cycles and concluded with a final extension step at 72°C.

This study was performed in line with the principles of the Declaration of Helsinki. Approval was granted by Saveetha Dental College & Hospitals (28/1/2022; Ref No. 22/perio/ 317). The participants were fully informed about the purpose, risks, and benefits of the study,

and their rights as participants. They signed a written informed consent form, which was obtained separately from each participant.

## 3. Results

The ICP4-KGP gingipain complex, which exhibited the lowest binding energy score of -288.29 kJ mol^-1, was identified as the most effective interaction, according to Hdock. Visualization of these interactions was performed using Pymol (Fig 1).

Fig 2 shows how the RMSD values of a protein-protein complex have changed over time. The complex reaches stability at 20,000 ps, as shown in the graph. Following that, the target's RMSD value fluctuations are under 1.5 Angstrom for the length of the simulation, which is entirely acceptable. After equilibration, the RMSD values of the ligand-protein fit to the receptor protein fluctuate within 1.0 Angstrom.

Peaks on the RMSF picture (Fig 3) indicate protein regions that fluctuated the most during the simulation. MD trajectories show higher-peaking residues in loop regions or N and C-terminal zones (Fig 4). Low RMSF values on binding site residues suggest stable protein-ligand binding.

Fig 4 shows the secondary protein structure elements spread by residue index. Beta strands are denoted by blue, while alpha helices are denoted by red.

Throughout the simulation, protein interactions with the ligand can be observed.

As shown in Figs 5 and 6, hydrogen bonds make up most of the crucial ligand-protein interactions that MD discovers. The exchanges and contacts depicted in Fig 5 are depicted as a timeline.

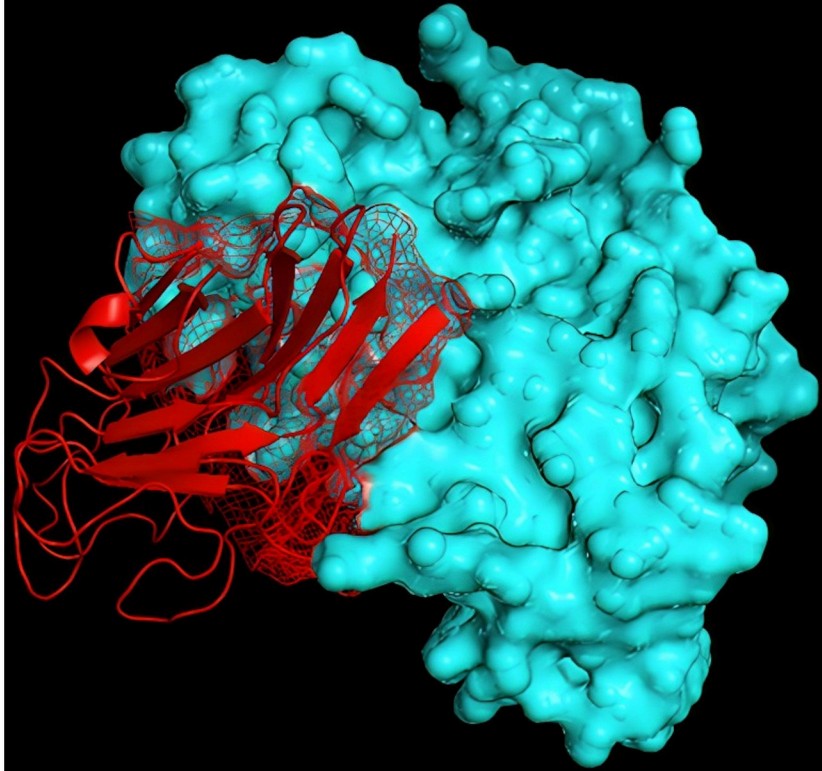

**Fig 1. Ligand 'KGP' interacts with receptor 'icp4' protein to form a binding relationship.** Ligands are shown in red, and receptor proteins are shown in cyan.

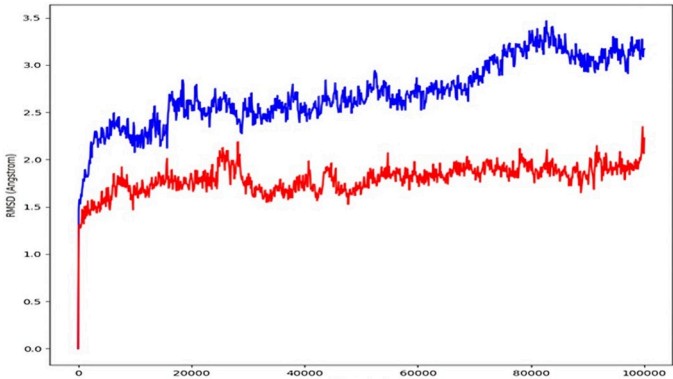

**Fig 2. The RMSD of receptor and ligand proteins varies over time, as indicated by the left Y-axis.** The RMSD of the receptor protein is blue, while the RMSD of the ligand protein is red.

The radius of gyration refers to how many atoms are distributed around a protein's axis (Rg). Rg and distance calculations are the two most essential indicators for predicting a macro-molecule's structural activity. Protein-ligand interactions with the radius of gyration changes show a conformational shift throughout time. A powerful computer could calculate the

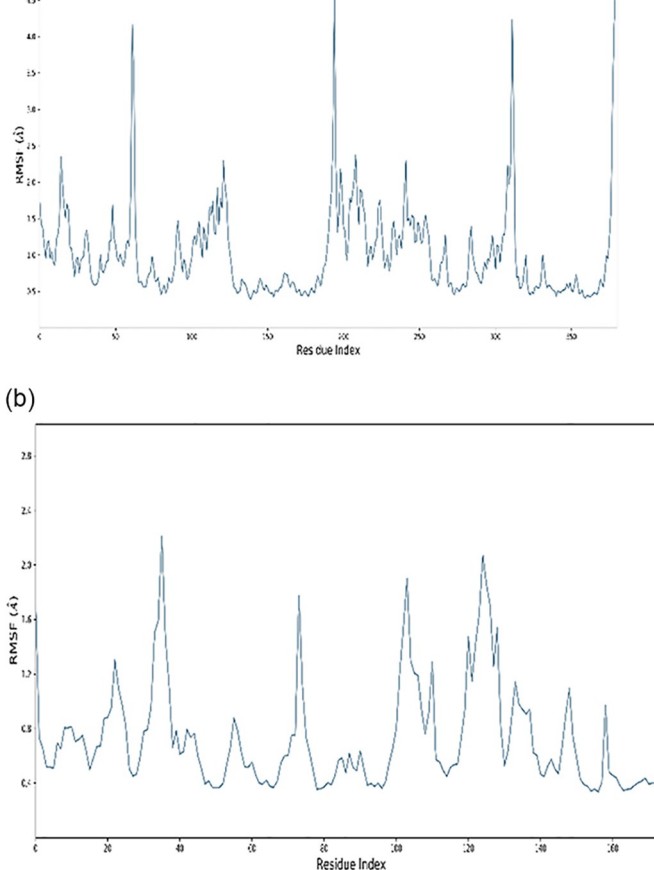

**Fig 3. A) Receptor protein, B) Ligand protein Residue-wise Root Mean Square Fluctuation (RMSF).**

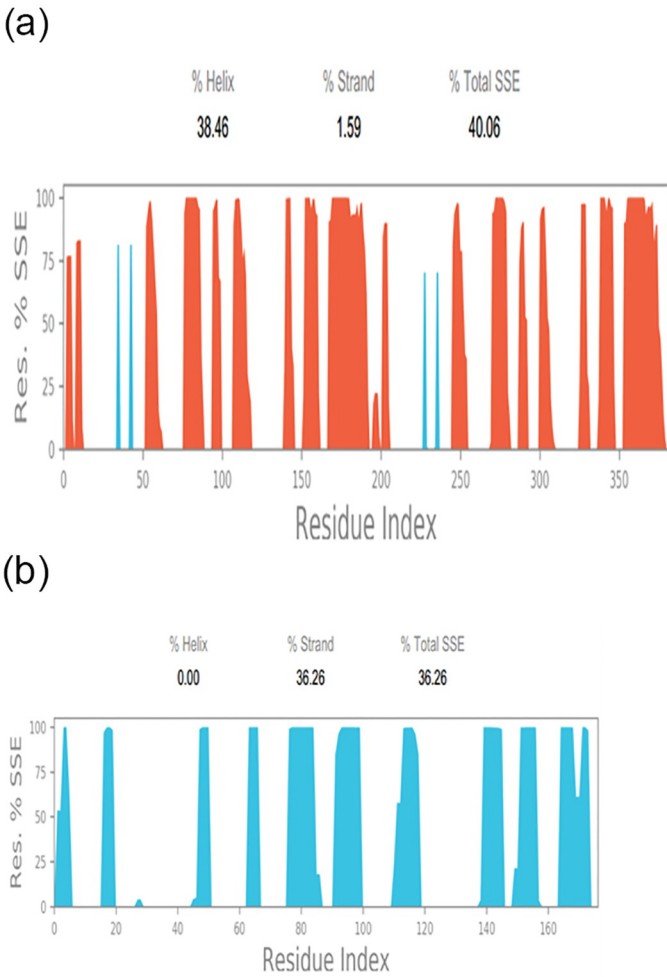

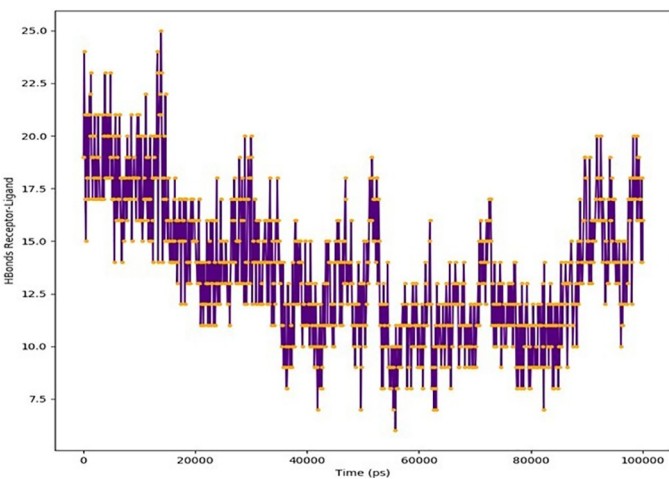

**Fig 4. Protein structure secondary structure elements spread by residue index.** Beta strands are denoted by blue, while alpha helices are denoted by red. A) The SSE of the receptor protein B) The SSE of the ligand-protein. The graph above shows the distribution of SSE across protein architectures by residue index.

**Fig 5. Interactions and contacts are depicted as timelines (H-bonds).**

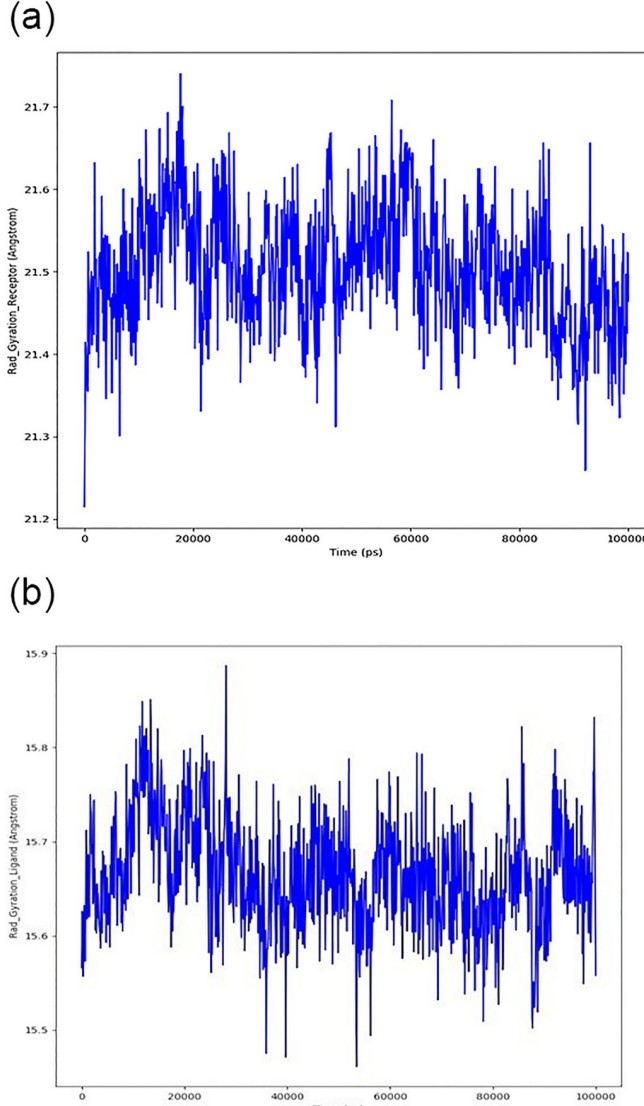

**Fig 6. Radius of gyration, A) Receptor Protein, B) Ligand Protein.**

gyration radius to determine a protein's compactness, which can be used to determine how quickly it folds.

## PCR results and agar gel electrophoresis

Fig 7 shows clinical validation of the presence of HSV micro-RNA 6 in periodontal tissue samples from periodontitis subjects using PCR. Thicker bands indicate higher hsv micro-RNA 6 in clinical samples.

## Discussion

Periodontitis is an inflammatory condition resulting from an imbalance in oral bacteria, triggering an exaggerated immune response and tissue damage supporting the teeth [12, 16]. The

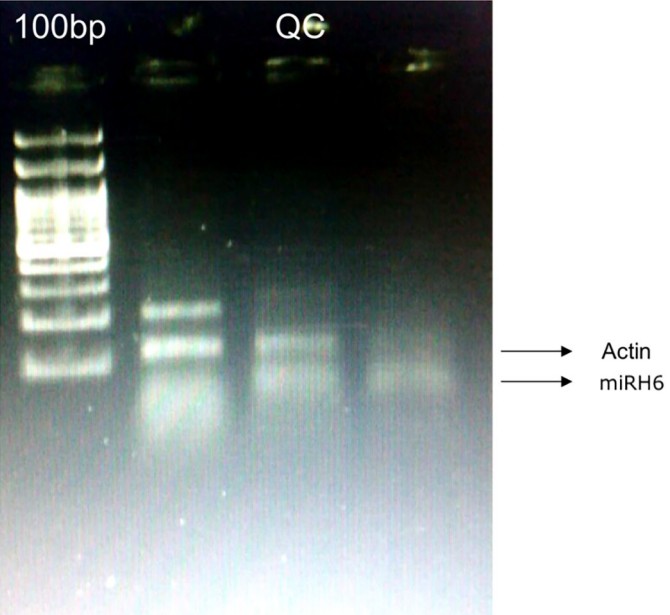

**Fig 7. Clinical validation of the presence of HSV micro-RNA 6 in periodontal tissue samples from periodontitis subjects using PCR.** Thicker bands indicate higher HSV micro-RNA 6 in clinical samples.

primary treatment involves cleaning the biofilm beneath the gum line and maintaining effective plaque control. Research indicates that regular periodontal maintenance therapy is crucial for sustaining post-initial progress [17, 18]. However, without consistent clinical monitoring and reinforcement of oral hygiene practices, these benefits may diminish, heightening the risk of recurrent periodontitis and tooth loss.

Good oral hygiene practices have been associated with a reduced prevalence of periodontitis. The principal cause of recurrent periodontitis is the buildup of dental plaque, a bacterial biofilm that forms on teeth. Recurrent periodontitis is characterized by an unknown etiology disorder marked by recurring severe inflammation of the periodontium, chronic and exacerbated immune responses, and resorption of tooth-supporting structures, including the alveolar bone, even after successful treatment [19].

Bacteria and viruses engage in interactions such as mutualism, parasitism, and predation, which play crucial roles in microbial populations, environmental dynamics, and human health. These interactions can significantly influence bacterial ecosystems, with bacteriophages regulating bacterial populations through infection and destruction of microorganisms. Altering the relative abundance of bacterial species can impact the composition and function of microbial communities.

Despite diligent efforts by dental professionals to treat periodontal disease, some individuals do not respond adequately [20]. These patients suffer from recurrent periodontitis due to resistance to treatment, likely influenced by microbial resistance and host immunological factors. HSV viral DNA infects tissues affected by periodontal disease, including gingival tissue, gingival crevicular fluid (GCF), and subgingival plaque. The latency property of HSV is particularly important, as it can be reactivated from neural ganglia in immunocompromised individuals, and its role in periodontitis remains unclear. This study represents the first to establish a connection between periodontitis and HSV latency [20–22].

The double-stranded DNA virus HSV-1 [23] can persist in periodontal epithelium and brain cells. HSV-1's replication cycle is driven by the regulatory gene ICP4, which transcribes both early and late viral genes. ICP4 interacts with micro-RNA to initiate HSV-1 reactivation from latency. A complex association exists between severe periodontal disease and herpesviruses such as EBV and HCMV. EBV, HCMV, and co-infections are commonly found in various periodontal disorders. Periodontitis patients often exhibit higher EBV DNA levels in saliva than healthy individuals. The prevalence of EBV in individuals with periodontitis correlates with pocket depth [23, 24], suggesting a connection between periodontitis, EBV, and periodontopathic bacteria.

Herpesviruses, particularly Epstein-Barr virus (EBV) [2], are implicated in serious gum disorders. While bacteria play a significant role in these conditions, antiviral medications may improve gum health by reducing EBV levels. Studies suggest that certain oral bacteria can activate EBV, contributing to inflammation and gum disease. There appears to be a detrimental cycle where bacteria and EBV exacerbate each other, worsening gum disease.

The periodontal bacterium *P. gingivalis* produces butyric acid, which can reactivate EBV [21, 23, 25]. Butyric acid in bacterial culture supernatants induces ZEBRA expression and histone H3 acetylation in EBV-infected cells, leading to their reactivation. *P. gingivalis* also induces histone acetylation and dissociation of HDAC from the BZLF1 promoter in latently infected cells, further facilitating EBV reactivation [8].

A microRNA (miRNA) is a short, non-coding RNA molecule composed of 18–22 nucleotide sequences found in plants, animals, and viruses. Upstream of the LAT promoter, miR-H1 and miR-H6 are located. Sequences upstream of the LAT promoter encode miR-H1, while miR-H6 encodes miR-H1. In HSV-1, miR-H6 targets ICP4 to help maintain latency. Except for miR-H6, all LAT-encoded v-miRs are oriented in the same direction as the LAT transcripts [24]. Unlike LAT, which is more prevalent in nerve ganglia in trigeminal neuralgia, miR-H6 maintains latency and activates productive disease activation. Evidence from several studies suggests a role for oral bacteria and HSV-miRNA in the development of periodontal inflammatory disorders.

Our study targeted icp4 transcripts interacting with the keystone pathogen *P. gingivalis*. According to George Hajishengallis's theory on "keystone pathogens," specific low-abundance microbial species can drive inflammation by disrupting the normal microbiome. Despite its low abundance, *P. gingivalis* can interact with ICP4 transcripts, influencing the periodontal inflammatory pathway.

Insilico docking enables fundamental investigations into protein interactions and provides a structural foundation for drug development. Protein-protein docking predicts the complex structures formed by individual protein molecules. The steric and physicochemical complementarity concept at the protein-protein interface is crucial for docking and simulation studies.

Insilico docking and simulation of protein-protein complexes revealed a strong inhibitory effect of *P. gingivalis* KGP gingipain with the ICP4 transcript, as evidenced by the docked complex achieving the lowest energy score of -288.29 kJ mol^-1. Throughout the simulation, the target's root mean square deviation (RMSD) values remained consistently below 1.5 Angstroms, indicating good stability [26]. Following equilibration, the RMSD of the ligand protein to the receptor protein was 1.0 Angstrom, demonstrating stable binding between the ligand and receptor proteins throughout the simulation. Compared to unstructured regions, alpha helices and beta strands exhibited firmer and more stable conformations, with higher peaks observed in the molecular dynamics (MD) trajectories primarily in loop areas or at the N- and C-terminal regions. Low RMSF (Root Mean Square Fluctuation) values of binding site residues further underscored the stability of the protein-ligand interaction [27–29].

Patients with recurrent inflammatory gingival and periodontal symptoms such as bleeding, halitosis, and pocket depths exceeding 7 mm at the same sites were selected for this study on periodontitis subjects. Analysis of patient samples from periodontitis cases revealed the presence of HSV microRNA H6, suggesting its role in activating HSV and potentially altering the periodontal microbiome to induce periodontitis [30]. This study highlights the role of *P. gingivalis*, a keystone pathogen [1, 2, 6], particularly in periodontitis, in potentially activating latent HSV in the periodontium by inhibiting ICP4 and suppressing HSV microRNA H6, thereby prolonging HSV infection and contributing to the development of periodontitis.

This study has several limitations, including a small sample size and the lack of a control group, which may limit the generalizability of the findings. Additionally, the absence of randomization could introduce selection bias. The use of insilico docking, while insightful, is inherently predictive and requires further validation through experimental studies. The study also focused on a specific subset of patients with recurrent inflammatory symptoms, potentially neglecting influences from host immune response, genetic predisposition, and lifestyle factors. These limitations should be addressed in future research to provide a more comprehensive understanding of the interplay between microbial and viral factors in periodontitis.

## Conclusion

This study provides preliminary evidence suggesting a potential link between *P. gingivalis*, herpes virus microRNA-6, and the pathogenesis of periodontitis. Insilico analysis demonstrated a plausible interaction between *P. gingivalis* KGP gingipain and ICP4, a critical viral regulatory protein. Furthermore, the consistent detection of miRNA-6 in periodontitis patients supports the hypothesis of viral involvement in disease progression. While these findings offer intriguing insights, the exploratory nature of this study necessitates further investigation with larger cohorts. A comprehensive understanding of the complex interplay between microbial and viral factors is essential for elucidating the aetiology of periodontitis and developing targeted therapeutic interventions.

## Supporting information

**S1 Checklist. STROBE statement—Checklist of items that should be included in reports of observational studies.**
(DOC)

**S1 File. Values used to build graphs.**
(DOCX)

**S1 Raw images.**
(PDF)

## Author Contributions

**Conceptualization:** Pradeep Kumar Yadalam, Raghavendra Vamsi Anegundi.

**Data curation:** Pradeep Kumar Yadalam.

**Formal analysis:** Pradeep Kumar Yadalam, Raghavendra Vamsi Anegundi, Carlos M. Ardila.

**Investigation:** Pradeep Kumar Yadalam, Anila Neelakandan, Rex Arunraj, Raghavendra Vamsi Anegundi, Carlos M. Ardila.

**Methodology:** Pradeep Kumar Yadalam, Anila Neelakandan, Rex Arunraj, Raghavendra Vamsi Anegundi, Carlos M. Ardila.

**Project administration:** Pradeep Kumar Yadalam.

**Resources:** Pradeep Kumar Yadalam.

**Supervision:** Pradeep Kumar Yadalam, Anila Neelakandan, Rex Arunraj, Raghavendra Vamsi Anegundi, Carlos M. Ardila.

**Validation:** Pradeep Kumar Yadalam, Anila Neelakandan, Rex Arunraj, Raghavendra Vamsi Anegundi, Carlos M. Ardila.

**Visualization:** Pradeep Kumar Yadalam, Anila Neelakandan, Rex Arunraj, Raghavendra Vamsi Anegundi, Carlos M. Ardila.

**Writing – original draft:** Pradeep Kumar Yadalam, Raghavendra Vamsi Anegundi, Carlos M. Ardila.

**Writing – review & editing:** Pradeep Kumar Yadalam, Carlos M. Ardila.

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
