## [Decision Letter · Decision Letter 0]

31 Jul 2024

PONE-D-24-24760Exploring the Interplay Between Porphyromonas gingivalis KGP Gingipain, Herpes Virus MicroRNA-6, and Icp4 Transcript in Periodontitis: Computational and Clinical InsightsPLOS ONE

Dear Dr. Ardila,

Thank you for submitting your manuscript to PLOS ONE. After careful consideration, we feel that it has merit but does not fully meet PLOS ONE’s publication criteria as it currently stands. Therefore, we invite you to submit a revised version of the manuscript that addresses the points raised during the review process.

I recommend revisiting the following points: Review the clinical part and explain in more detailSpecify whether the periodontal therapy is surgical or SRP, it is not clear in the textThe conclusion is wrong because herpes virus is not present in all patients with periodontitis. Review the conclusionsReview some typos (Insilco instead of In silico)How the tissue collection was done and the region of collection should be better described. It was not a byopsy but a scaling removal. What type of tissue was achieved? Inclusion and excluison criteria also needs clarification. Include clinical signs of sites "recurrent"Results: The atuhros could better work on Figures. SOme of them can be merged and give the reader a easy view of in silico resutFocus the conclusions more on the results obtained

Overall the work is good and the topic interesting

We look forward to receiving your revised manuscript.

Kind regards,

Maria Giulia Nosotti, Master's Degree

Academic Editor

PLOS ONE

Journal Requirements:

Reviewers' comments:

Reviewer's Responses to Questions

**Comments to the Author**

1. Is the manuscript technically sound, and do the data support the conclusions?

Reviewer #1: Yes

Reviewer #2: Partly

2. Has the statistical analysis been performed appropriately and rigorously? 

Reviewer #1: Yes

Reviewer #2: N/A

3. Have the authors made all data underlying the findings in their manuscript fully available?

Reviewer #1: Yes

Reviewer #2: Yes

4. Is the manuscript presented in an intelligible fashion and written in standard English?

Reviewer #1: Yes

Reviewer #2: Yes

5. Review Comments to the Author

Reviewer #1: The study "Exploring the Interplay Between Porphyromonas gingivalis KGP Gingipain, Herpes Virus MicroRNA-6, and Icp4 Transcript in Periodontitis: Computational and Clinical Insights " is an interesting analysis of multiples approaches in subjects affected by periodontitis. Results indicate a relatinhsip between Pg and HSV in periodontitis occurence. There are some points to be clarified before further analysis.

Abstract: The authors can review Background and include computational analysis in Methods. As presetned, it is not possible understand both analysis.

Introduction: Authors did a good job. The text is clear and explain the hypothesis and the plausibility of the study.

Methods

Review some typos (Insilco instead of In silico)

How the tissue collection was done and the region of collection should be better described. It was not a byopsy but a scaling removal. What type of tissue was achieved? Inclusion and excluison criteria also needs clarification. Include clinical signs of sites "recurrent".

Results: The atuhros could better work on Figures. SOme of them can be merged and give the reader a easy view of in silico resut. A Graphical abstrct of findigns could be an interesting option to explain this new pathway.

Conclusion could be less especulative and focus on results.

Reviewer #2: Dear Authors

The clinical part of the manuscript is very confusing. Since there is no equivalent for recurrent periodontitis and the new classification the stud patients can't be assigned automatically to Periodontitis Stage 3 (line 164-166) if they do not fit the requirements. Also the classification is made base on CAL and not PD so you need to correct this since the only way to precisely asses the bone loss is through CAL calculation. It is also not clear what kind of therapy the patients received. In a part of the manuscript you state they received surgical therapy then in other SRP and in other conventional therapy. You must be more specific with that and stick to it through all the manuscript.

Line 168-169 it is not clear what you meant here since there is no timepoint description in your manuscript so T1 sites has no meaning (assuming T is from timepoint).

The study has no limitations written even if in STROBE declaration they are listed as written in page 11.

Also the conclusion is wrong because herpes virus is not present in all patients with periodontitis. This is why it was important to classify correctly their patients.

Also in order to provide a clinical validation you must do this by evaluating clinical parameters. What you have done is a PCR validation of the computational analysis.

6. PLOS authors have the option to publish the peer review history of their article (what does this mean?). If published, this will include your full peer review and any attached files.

Reviewer #1: No

Reviewer #2: No

---

## [Author Response · Author response to Decision Letter 0]

5 Sep 2024

Dear Editor and Referees, 

We are grateful for the constructive comments you provided, which helped us to improve the manuscript significantly. 

Our responses to your comments are outlined below and highlighted in yellow in the new version.

Editor

Thank you for your constructive feedback and recommendations. We have carefully reviewed the points you raised and addressed them in detail in our responses to the reviewers' comments.

Reviewer #1: 

1. Abstract: The authors can review Background and include computational analysis in Methods. As presented, it is not possible understand both analysis.

RESPONSE: Thank you for your valuable feedback. We have reviewed and refined the Background section of the abstract for clarity and conciseness. Additionally, we have revised the abstract to include the computational analysis within the Methods section. The updated abstract now clearly describes the computational docking analysis performed to investigate the interaction between ICP4 and KGP gingipain, alongside the clinical methods used to detect HSV microRNA-6. We hope this revision meets your expectations and provides a clearer understanding of both analyses conducted in the study.

2. Methods.

Review some typos (Insilco instead of In silico).

RESPONSE: The typos were corrected.

How the tissue collection was done, and the region of collection should be better described. It was not a biopsy but a scaling removal. What type of tissue was achieved? Inclusion and exclusion criteria also need clarification. Include clinical signs of sites "recurrent".

RESPONSE: All these aspects were resolved in the revised version.

3. Results: The authors could better work on Figures. Some of them can be merged and give the reader an easy view of in silico result. A Graphical abstract of findigns could be an interesting option to explain this new pathway.

RESPONSE: The figures were corrected.

4. Conclusion could be less speculative and focus on results.

RESPONSE: In the revised version, the conclusions focus on the results obtained.

Reviewer #2: 

1.The clinical part of the manuscript is very confusing. Since there is no equivalent for recurrent periodontitis and the new classification the study patients can't be assigned automatically to Periodontitis Stage 3 (line 164-166) if they do not fit the requirements. Also, the classification is made based on CAL and not PD, so you need to correct this since the only way to precisely assess the bone loss is through CAL calculation. 

RESPONSE: Thank you for your valuable feedback. We have used both probing depth and clinical attachment loss for assessing the severity.

2. It is also not clear what kind of therapy the patients received. In a part of the manuscript, you state they received surgical therapy then in other SRP and in other conventional therapy. You must be more specific with that and stick to it through all the manuscript.

RESPONSE: Thank you for your valuable feedback. All patients received non-surgical therapy followed by surgical therapy as a standard protocol.

3. Lines 168-169 it is not clear what you meant here since there is no timepoint description in your manuscript so T1 sites has no meaning (assuming T is from timepoint).

RESPONSE: The recommendation was amended in the revised version.

4. The study has no limitations written even if in STROBE declaration they are listed as written in page 11.

RESPONSE: The limitations were added at the end of the discussion.

5. Also, the conclusion is wrong because herpes virus is not present in all patients with periodontitis. This is why it was important to classify correctly their patients.

RESPONSE: The recommendation was amended in the revised version.

6. Also, to provide a clinical validation, you must do this by evaluating clinical parameters. What you have done is a PCR validation of the computational analysis.

RESPONSE: Thank you for your insightful comment. We acknowledge the importance of clinical validation in evaluating the clinical parameters of our study. We consider that the insilco validation of the P. gingivalis-ICP4 interaction is a crucial step in confirming the interaction between MIRH-6 transcription factors and ICP4 in maintaining latency in the nerve ganglia of the periodontium. This interaction is fundamental to understanding the role of P. gingivalis gingipain KGP and its inhibition of ICP4 in the development and progression of periodontal disease.

Protein-protein docking, and molecular dynamics simulation are necessary to assess the stability and dynamics of the protein-protein interactions between KGP and ICP4. These computational techniques provide insights into the binding affinity and structural changes of the complex, helping to validate the proposed interaction and understand its functional implications.

The clinical validation study is important for providing evidence of the relevance of the P. gingivalis-ICP4 interaction in human periodontal disease. By selecting patients who have demonstrated additional clinical attachment loss despite standard periodontal treatments, the study aims to show the association between this interaction and treatment-resistant or recurrent periodontal disease.

The inclusion and exclusion criteria are carefully chosen to ensure the study population accurately represents the target patient group. The thorough monitoring and collection of tissue samples allow for the analysis of HSV micro-RNA and its potential role in the disease process.

Overall, the insilco validation and clinical validation study provide important scientific and clinical evidence to support our hypothesis and contribute to a better understanding of the mechanisms underlying periodontal disease.

Please see the next references:

Manaithiya A, et al. Elucidating molecular mechanism and chemical space of chalcones through biological networks and machine learning approaches. Comput Struct Biotechnol J. 2024 Jul 6;23:2811-2836. doi: 10.1016/j.csbj.2024.07.006. 

Brueckner AC, et al. MDFit: automated molecular simulations workflow enables high throughput assessment of ligands-protein dynamics. J Comput Aided Mol Des. 2024 Jul 17;38(1):24. doi: 10.1007/s10822-024-00564-2. 

De La Torre S, et al. Computational approaches for lead compound discovery in dipeptidyl peptidase-4 inhibition using machine learning and molecular dynamics techniques. Comput Biol Chem. 2024 Jul 10;112:108145. doi: 10.1016/j.compbiolchem.2024.108145.

---

## [Decision Letter · Decision Letter 1]

2 Oct 2024

Exploring the Interplay Between Porphyromonas gingivalis KGP Gingipain, Herpes Virus MicroRNA-6, and Icp4 Transcript in Periodontitis: Computational and Clinical Insights

PONE-D-24-24760R1

Dear Dr. Ardilla,

We’re pleased to inform you that your manuscript has been judged scientifically suitable for publication and will be formally accepted for publication once it meets all outstanding technical requirements.

Kind regards,

Maria Giulia Nosotti, Master's Degree

Academic Editor

PLOS ONE

Additional Editor Comments (optional):

Reviewers' comments:

Reviewer's Responses to Questions

**Comments to the Author**

1. If the authors have adequately addressed your comments raised in a previous round of review and you feel that this manuscript is now acceptable for publication, you may indicate that here to bypass the “Comments to the Author” section, enter your conflict of interest statement in the “Confidential to Editor” section, and submit your "Accept" recommendation.

Reviewer #2: All comments have been addressed

2. Is the manuscript technically sound, and do the data support the conclusions?

Reviewer #2: Yes

3. Has the statistical analysis been performed appropriately and rigorously? 

Reviewer #2: N/A

4. Have the authors made all data underlying the findings in their manuscript fully available?

Reviewer #2: Yes

5. Is the manuscript presented in an intelligible fashion and written in standard English?

Reviewer #2: Yes

6. Review Comments to the Author

Reviewer #2: In the abstract and manuscript in the conclusion please modify in the last sentence "periodontitis" with "reccurent periodontitis". It is more accurate since all your patients had recurrent periodontitis.

7. PLOS authors have the option to publish the peer review history of their article (what does this mean?). If published, this will include your full peer review and any attached files.

Reviewer #2: No

---

## [Editor Report · Acceptance letter]

7 Oct 2024

PONE-D-24-24760R1 

PLOS ONE

Dear Dr. Ardila, 

I'm pleased to inform you that your manuscript has been deemed suitable for publication in PLOS ONE. Congratulations! Your manuscript is now being handed over to our production team.

Kind regards, 

on behalf of

Dr. Maria Giulia Nosotti 

Academic Editor

PLOS ONE